# Assessing the Genomics Structure of Dorper and White Dorper Variants, and Dorper Populations in South Africa and Hungary

**DOI:** 10.3390/biology12030386

**Published:** 2023-02-28

**Authors:** George Wanjala, Putri Kusuma Astuti, Zoltán Bagi, Nelly Kichamu, Péter Strausz, Szilvia Kusza

**Affiliations:** 1Centre for Agricultural Genomics and Biotechnology, Faculty of Agricultural and Food Sciences and Environmental Management, University of Debrecen, 4032 Debrecen, Hungary; 2Doctoral School of Animal Science, University of Debrecen, 4032 Debrecen, Hungary; 3Directorate of Livestock Production, Bungoma P.O. Box 437-50200, Kenya; 4Ministry of Agriculture Livestock, Fisheries and Cooperatives, State Department of Livestock Development, Naivasha Sheep and Goats Breeding Station, Naivasha P.O. Box 2238-20117, Kenya; 5Institute of Strategy and Management, Corvinus University of Budapest, 1093 Budapest, Hungary

**Keywords:** adaptation, dorper, genetic diversity, linkage disequilibrium, management best practices, population structure

## Abstract

**Simple Summary:**

The Dorper sheep breed was created to thrive in harsh environments in South Africa. Two breed variants were developed in the selection process. The breed gained popularity and was exported to several regions of the world where it is reported to be doing better than the purported locally adapted sheep breeds. To enhance the performance of other native sheep breeds, Dorper is widely utilized in crossbreeding with native breeds. There has not been any research done to examine the genomic status of Dorper in South Africa and Dorper in other places outside of South Africa. The genomic architecture of white Dorper and Dorper from South Africa and Hungary are compared in this study. White Dorper, Dorper, and Dorpers from South Africa and Hungary all have significantly distinct genomes. Different environmental factors and variations in coat color could be the cause of the genetic variations.

**Abstract:**

Dorper sheep was developed for meat production in arid and semi-arid regions under extensive production systems in South Africa. Two variants with distinct head and neck colors were bred during their development process. White Dorper have a white coat while Dorper have a black head and neck. Both variants have grown in popularity around the world. Therefore, understanding the genomic architecture between South African Dorpers and Dorper populations adapted to other climatic regions, as well as genomic differences between Dorper and White Dorper variants is vital for their molecular management. Using the ovine 50K SNP chip, this study compared the genetic architecture of Dorper variants between populations from South Africa and Hungary. The Dorper populations in both countries had high genetic diversity levels, although Dorper in Hungary showed high levels of inbreeding. White Dorpers from both countries were genetically closely related, while Dorpers were distantly related according to principal component analysis and neighbor-joining tree. Additionally, whereas all groups displayed unique selection signatures for local adaptation, Dorpers from Hungary had a similar linkage disequilibrium decay. Environmental differences and color may have influenced the genetic differentiation between the Dorpers. For their molecular management and prospective genomic selection, it is crucial to understand the Dorper sheep’s genomic architecture, and the results of this study can be interpreted as a step in this direction.

## 1. Introduction

The hardy South African composite breed Dorper was created in the 1930s by mating the Black-headed Persian and the Dorset Horn. The primary breeding goal of the Dorper breed’s founders was to develop a breed of sheep suitable for producing slaughter lambs in arid and semi-arid environments under an extensive production system. Therefore, Dorper’s founders aimed to combine the Dorset-Horn’s capacity to produce mutton with the Black-headed Persian’s hardiness [1]. Two Dorper variants (Dorper and White Dorper) were created through two distinct selection processes [2,3]. It is suggested that both Dorper variants possess excellent growth [4], meat quality [5], and reproductive traits [1,6]. The Dorper breed is increasingly getting popular worldwide, not just in South Africa.

The breed is used in crossbreeding with native sheep in several countries to improve the performance of the offspring. For instance, Dorper×Red Maasai lambs were observed to grow at higher rates than the average of their pure parents in Kenya [7,8], the Dorper×Santa Inês breed was observed to grow at a higher rate in Brazil [9], and the Dorper×Chinese Mongolian sheep breed grew at a higher rate and produced better carcasses than the parental [10]. In Europe, Dorper crossbreeds have also recorded better performance than the average of their parents, e.g., in comparison to the Romanian Turcana native sheep breed. Gavojdian et al. [11] reported higher growth and reproductive performance in Dorper×Turcana offspring. Awassy Co. Farm first introduced the Dorper breed to Hungary in 2006 to improve the quantity and quality of the extensively reared Gyimesi Racka breed’s meat without changing the existing sheep management system [6]. Another batch of Dorper and White Dorper from France and Switzerland, respectively, were introduced at the University of Debrecen in 2008 [12]. Studies on the performance of Dorper×several indigenous sheep breeds in Hungary showed improved growth performance of crossbred lambs [6].

The relationship between genotype and phenotype as well as the interaction between genotype and environment are of great interest at present. According to previous research, the environment has a significant impact on how an organism’s genetic architecture develops. While going through their process of natural selection, the organism goes through “genomic evolution” to adapt to the environment. Following domestication, the interaction between genotype and environment altered the ovine species as they inhabited diverse ecological regions and evolved into numerous breeds with varying phenotypic expressions. Recently, numerous studies reported breed eco-regional genetic variation, e.g., [13,14,15]. The Dorper breed developed and adapted to the South African sub-tropical climate and might have undergone genomic changes to adapt to the new environments they are exported to. In addition, the White Dorper and Dorper have the same ancestry and share many traits, although they have different coat colors. While White Dorper has a white covering, Dorper has a black head and the top part of the neck, but the body and legs are white. A recent study on the molecular basis of a similar breed with varying color variants, e.g., Hungarian white and black Racka, has shown non-negligible levels of differentiation [16]. The genetic differences between the original Dorper from South Africa and the Dorper populations outside of South Africa have not been studied. Additionally, no research has been done on the genomic variations between the White Dorper and Dorper variants. Therefore, a better opportunity for their molecular management would result from understanding the genomic status of these Dorper and White Dorper populations.

We contrasted the genome architecture of the Dorper populations found in South Africa and Hungary because we hypothesize that the latter may have undergone genomic evolution to survive the new temperate climate rather than the tropical climate in which they were originally developed. Therefore, our objectives were to (i) examine the genetic position of Dorpers’ within the world sheep population, (ii) assess the genomic status of both Dorper populations from Hungary and South Africa, and (iii) identify and contrast the selection signatures for each population. Alongside it, we present our well-considered view on how Africa and other developing countries might enhance the management of their domestic animal genetic resources.

## 2. Materials and Methods

To represent breed-based populations, samples from unrelated individuals for each breed were gathered from several farms. Vacutainer tubes were used to collect blood samples for Hungarian populations (Dorper, White Dorper, and Hortobágy Racka breeds) from the jugular vein. The total number of samples collected was *n* = 20 for the Dorper (HUDOR), *n* = 20 for the White Dorper (HUWDOR), and *n* = 20 for the Hortobágy Racka (HORAC) breed, which served as an out-group. This study also included South African Dorper (SADOR; *n* = 21) and South African White Dorper (SAWDOR; *n* = 5), which were retrieved from the WIDDE [17] database and were part of the breeds analyzed by Kijas et al. [18]. SADOR and SAWDOR will be referred to as the “South African populations/samples” in this study, whereas HUDOR, HUWDOR, and HORAC will be referred to as the “Hungarian populations/samples”.

DNA extraction and genotyping of the Hungarian samples were done by Neogen Company. Ovine50K Illumina microarray bead chips [19] were produced based on Oar v4.0 as the reference genome. We received genomic datasets from the final report from which the working datasets of Hungarian samples were generated. A dataset known as “MergedBeforeQC” was created by merging the South African and Hungarian samples. Only SNPs and samples that passed the following threshold in Plink 1.09 [20] were included in the downstream analysis; autosome SNPs, individuals with not more than 10% missing data, SNPs with minor allele frequency (MAF) of not more than 5%, and all SNPs that met the hardy Weinberg equilibrium threshold of 0.1 × 10^−6^.

The Arlequin software version 3.5 [21] was used to evaluate the population genetic diversity indicators and the relationship between populations. Only populations with more than fifteen (15) samples were included in the analysis. Hence, SAWDOR was excluded. Expected heterozygosity (He) and observed heterozygosity (Ho) are the diversity indices calculated. The same data set was used in Plink 1.09 [20] to compute the moment relatedness F coefficient, which is used to estimate the distribution of inbreeding and relatedness among and between individuals in the population [22] in plink 1.09. The detectRUN package [23], which is implemented in the R environment [24] was used to identify the Runs of Homozygosity (ROH) using the sliding window method. All studied populations were included in the analysis. The following criteria and thresholds were used to define the ROH: a sliding window of 50 SNPs, a maximum missingness of 1 allele, a minimum length of 25,000 bp, at least 30 homozygous SNPs included in the ROH, a minimum density of 1 SNP per 1000 kilobases, and a minimum length of 1 Mb. The algorithm of ROH is extensively described by Xu et al. [25].

Plink 1.09 software was used to perform principal component (PC) analysis on two separate datasets. The first dataset, dubbed “MergedGlobalOvineDataset,” was created by merging “MergedBeforeQC” with global sheep datasets examined by Kijas et al. [18]. The goal of the PCA of the “MergedGlobalOvineDataset,” dataset was to determine the genomic position of the Dorper breed globally. Only SNPs and individuals that passed the quality threshold as described for “MergedAfterQC” were included in the PC computation. The second dataset used for PC computation was “MergedAfterQC”. The aim was to establish the genomic distance between four Dorper populations, i.e., HUDOR, HUWDOR, SADOR, and SAWDOR, with HORAC as an outgroup. Visualization of the PC plots was done by the ggplot 2 package [26] implemented in the R environment [24].

Further, autosomes of only four Dorper populations were used to perform an Analysis of Molecular Variance (AMOVA) and fixation index (FST) computation using the Arlequin 3.5 software [21]. The fixation index (FST) was estimated to establish population divergence. The neighbor-joining tree was performed on “MergedAfterQC” by Trait Analysis with the aSSociation, Evolution, and Linkage (TASSEL) v5.0 software [27].

The linkage disequilibrium (r^2^) matrix was calculated in Plink 1.09 [20] using the following criteria: a maximum window size of 100 SNPs and a maximum distance between the SNPs of 1000 kbs. The data quality thresholds described before were performed on each population separately. The LD plot per breed was visualized in the ggplot2 package [26] implemented in R software [24].

Using the default thresholds, the rehh package [28] implemented in R software [24] was used to perform a genome-wide scan of each population of interest independently to search for selection signatures for local adaptation using an integrated haplotype score (iHS). The iHS statistics compare the alleles within the population. Visualization of Manhattan plots was generated by the qqman package [29] implemented in R software and R studio. The threshold of the top 0.001 was used to designate the genomic regions of interest (top hits regions based on −log10P). Using the BiomaRt package [30] in the R environment, protein-coding genes that are close to the genomic regions of interest and span about ±250,000 kbps were extracted from the Ensemble [31] database.

## 3. Results

### 3.1. Basic Population Statistics and Runs of Homozygosity

Basic population statistics and ROH are shown in Table 1. High levels of both He and Ho were observed in all the analyzed groups. On the one hand, SADOR (0.372 ± 0.164) had the highest values of Ho, while HUDOR (0.365 ± 0.177) had the lowest among the Dorper breeds. The HUDOR on the other hand had the lowest He (0.351 ± 0.145), whilst the SADOR had the highest (0.405 ± 0.128). Both He and Ho in all the studied populations had a relatively high standard deviation of >0.1. Another important point to note is that all Dorper populations had a higher Ho than He. In terms of the inbreeding coefficient (F), HUDOR had the highest (0.173 ± 0.1106). Moreover, the largest percentage of both short (0–6 mbps) and long (>48 mbps) ROH segments were observed in HUDOR and HUWDOR.

### 3.2. Population Structure and Differentiation between Populations

Dorpers from both Hungary and South Africa are highly structured, with only 9.76% of variation being explained by variation between populations and more than 90% being explained by variation within groups. Given that the FST value is 0.097 (*p* < 0.001), the populations under study are significantly different (Table 2).

In addition, on a PC plot representing the global sheep populations, Dorper populations from both South Africa and Hungary are grouped together. Nevertheless, there was no discernible overlap between South African and Hungary’s populations. On the same graph, all Dorpers were clustered away from the Racka group (Figure 1A). While White Dorpers clustered together without overlapping into either group, SADOR and HUDOR formed separate clusters in a Principal Component plot of the five populations in the present study (Figure 1B). Notably, the White Dorpers had a genomic position between SADOR and HUDOR (Figure 1B). Four primary, different branches of SADOR, HUDOR, HORAC, and White Dorpers were visible in the neighbor-joining tree (Figure 1C). Two sub-branches of the White Dorpers branch separated the populations of HUWDOR and SAWDOR.

### 3.3. Linkage Disequilibrium (LD) and Signatures of Selection for Local Adaptation

The identified entire genome selection sweeps vary between populations (Appendix A). Although more than 3000 genomic areas were shown to be possibly under selection, very few of them had annotated genes. The top hit regions of interest were also observed in different chromosomes in each population (Figure 2A–E).

With increasing marker distance, all groups showed a decreasing linkage disequilibrium (Figure 2F). In Dorpers, the rate of LD decline appeared to be comparable. However, in Hortobágy Racka, the rate of LD decline appeared to be faster than that of Dorpers’ up to a maximum of 250 Mbs. While the SAWDOR and SADOR had different LD values, HUDOR and HUWDOR appeared to have similar LD values throughout the genome. When compared to the other populations in the study, SAWDOR exhibited the largest mean LD values (SADOR LD = 0.365 ± 0.156, SAWDOR LD = 0.564 ± 0.251, HUDOR = 0.420 ± 0.199, HUWDOR = 0.421 ± 0.200, and HORAC LD = 0.338 ± 0.141).

## 4. Discussion

### 4.1. Basic Population Statistics and Runs of Homozygosity (ROH)

High levels of diversity were present in all Dorper populations from both countries, with observed heterozygosity varying from 0.372 ± 0.164 to 0.365 ± 0.177. The high standard deviation in Dorper populations could be due to high genetic variation within the populations. Notably and expectedly, Ho scores in all populations except HORAC were higher than He possibly due to the selection process the breed has undergone since its development. The basis of evolution is a significant genetic variation within populations. Breeds with a wide genetic variability stand a good chance of adapting to the effects of climate change. As a result, Dorper has a high likelihood of adapting to a variety of environmental situations given the genetic diversity that currently exists within populations. This explains its popularity and successful performances in regions other than South Africa. The current findings support many other studies that found that indigenous sheep breeds are genetically diverse, e.g., [32,33,34,35].

The moment of relatedness F coefficient observed in the current investigation showed that the SADOR and SAWDOR populations are less inbred than the HUDOR and HUWDOR. This was expected given that there are only about 200 Dorpers and White Dorpers in all of Hungary, compared to over 48,000 Dorpers and White Dorpers in South Africa [36] and thus HUDOR and HUWDOR are classified as being at risk of extinction.

Both HUDOR and HUWDOR populations had the highest number of short (0–6 mbps) and long (>48 mbps) ROHs. Two copies of a haplotype from a common ancestor combine in an individual to form ROH. Short runs occur from haplotypes inherited from distant ancestors, while long runs are indicative of haplotypes inherited from a recent common ancestor [37].

Since local recombination and mutation rates as well as other evolutionary processes have a significant role, the pattern and distribution of ROHs along the genome are not random or homogeneous. As a result, the development and distribution of ROHs across the genome are the results of the interaction between various factors, including recombination rate and selection pressure [38]. In the present study, longer ROH (>48 mbps) agrees with the results of the inbreeding coefficient in both Dorper populations in Hungary. The patterns of ROH per breed are available in Appendix A, which shows a different pattern and distribution among breeds.

### 4.2. Population Structure and Differentiation between Populations

In comparison with the global sheep dataset analyzed by Kijas et al. [18], the principal component plots show that Dorpers comprise a distinct cluster (Figure 1A). Unexpectedly, only Dorpers and Hortobágy Racka’s independent PC plot indicated that HUDOR and SADOR are as distantly related between themselves as they are to the White Dorpers. Moreover, HUWDOR and SAWDOR were shown to be closely related on the same PC plot (Figure 1B). These results were supported by the neighbor-joining tree (Figure 1C). While the distinction between the examined populations can be attributed to recent selection pressure and environmental influence, the distinction by color variant can also be attributed to genetic variations underlying the color. The study by Zsolnai et al. [16], in which Racka sheep clustered based on color variants, and the study by Blackburn et al. [14], which also demonstrated that Hereford beef cattle clustered based on where the samples were obtained are in agreement with the current findings. According to the Analysis of Molecular Variance (AMOVA; Table 2), a greater fraction of population variation results from individual variation within a single population, providing a chance for within-population selection. Although the variation between populations is small (9.6%), it is considerable and high when considering the histories of the Dorper populations under study. Additionally, pairwise FST was 0.097 (*p* < 0.001), suggesting a non-negligible differentiation. The FST values found in the current study are higher than those found in Tunisian [39] and Moroccan [40] sheep breeds [40] as well as in some of the Hungarian breeds reported in Kusza et al. [41].

### 4.3. Linkage Disequilibrium (LD) and Signatures of Selection for Local Adaptation

It was found that when marker distance increases, linkage disequilibrium diminishes in all groups (Figure 2F). For the SAWDOR, HUWDOR, SADOR, and HORAC, at a distance of 2000 kb, the r^2^ values were approximately 0.5, 0.3, 0.3, 0.25, and 0.2, respectively. The r^2^ values observed in this study are comparable to those obtained in Spanish Churra sheep [42] and Iranian sheep [43] which are higher than those reported in Sicilian dairy sheep breeds [44]. Shifman [45] proposed that the r^2^ of at least 0.3 is a significant value for QTL mapping. The rate of LD decay is vital in the implementation of genomic selection, genome-wide association studies, as well as defining the population recombination history.

Notably, the number of genes found in each population varied and were located on several chromosomes. For instance, the HUDOR, HUWDOR, SADOR, SAWDOR, and HORAC, respectively, had around 5, 13, 12, 7, and 22 genes mapped out (See Appendix A for more details).

In HUDOR, the mapped genes included Uncoupling Protein 2 (UCP2) which is believed to be responsible for the control of body temperature and regulation of energy balance [46]. Immune response-related genes mapped in the same population included Signal Transducer and Activator of Transcription 1 (STAT1) and Interleukin-7 (IL7). In HUWDOR, out of thirteen genes mapped, eight were believed to be underpinning responses to immunity [47]. Among others, they include Fatty Acid Binding Protein 5 (GUCA2A), Interleukin-2 (IL-2), Fibroblast Growth Factor 2 (FGF2), and Interleukin 21 (IL-21). Similarly, in SAWDOR, five genes: Fc Gamma Receptor IIIa (FCGR3A), Fc Gamma Receptor II a (FCGR2B), Superoxide Dismutase 1 (SOD1), and Interleukin-7 (IL-7) were the genes believed to likely be underpinning immunity response, whereas MX Dynamin Like GTPase 2 (MX2), MX Dynamin Like GTPase 1 (MX1), Adenosine A3 Receptor (ADORA3), and Interleukin 21 Receptor (IL-21) were believed to be controlling the immune response in the SADOR population. It is important to note here that immune response is not the only main adaptation trait for many livestock species. Adaptation is a multi-factorial trait that involves a combination of multiple physiological mechanisms within an individual. In the present paper, we mapped many genes that perform different functions supporting the preceding statement. However, in reaction to climate change, the immunological response has emerged as one of the traits that animals are claimed to have acquired in order to adapt to the local environmental conditions, particularly when they are raised in a natural habitat (extensive production system).

Gene names and their proposed functions were retrieved from the GeneCard [47] database. Though more than 3000 genomic areas were shown to be under selection, only a few of these regions contain names of annotated genes. This may be possible given that the sheep breeds used to build the reference genome may have originated in areas with different climatic conditions than the populations being studied. To improve the precision of identifying the signatures of local adaptation or implementation of genome-wide association studies, breed-specific reference, and/or regional-based reference genomes are therefore required.

## 5. Potential Possibilities for Improving the Genomic Management of Animal Genetic Re-Sources in Developing Nations

All parties involved must work together to address the issue of inadequate breeding strategies and research in Africa and other developing countries that are impeding the adoption of novel genomic applications. To assist in the application of these technologies, benchmarking from regions such as the European Union, Great Britain, and the USA that has already achieved success in the management of animal genetic resources can be done. It is important to close the information flow gap and bridge the gap between researchers and end users. Through their advisory activities, such as the advisory and other services provided by the chambers of agriculture in the European Union [48], government organizations, farmers’ associations, and other important stakeholders could play a significant role in distributing information and assisting breeders.

To inform farmers and breeders of the advantages of genomic applications and how to correctly use and interpret the data, capacity development and training programs might be set up. To ensure the greatest comprehension and uptake, these programs might be customized to the local context and language. Additionally, public-private collaborations might be created to advance the study and creation of regionally tailored genomic technologies and to make them available and affordable. Government agencies could offer financial help and policy measures, while private enterprises could offer technical aid.

## 6. Conclusions

The findings of the current study show that Dorper populations are very genetically diverse, with a significant amount of variation originating within the population. The genomic differences observed may be due to different environment-based selection pressures that each population experiences. Thus, the genomic differences between Dorper populations in South Africa and Hungary may have been influenced by environmental factors. Even after going through a similar developmental process, the Dorper and White Dorper variants differ significantly in their genomic makeup possibly due to the continued and long-term selection differences. Given the popularity of Dorper populations worldwide, the findings of this study provide a foundation for their molecular management.

## Figures and Tables

**Figure 1 biology-12-00386-f001:**
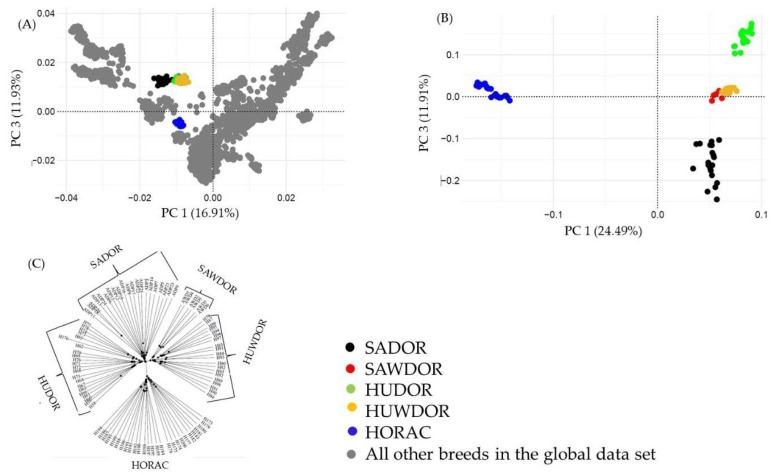
Population clustering by PCA and neighbor-joining tree. (**A**) Principal component for global sheep populations. (**B**) Principal component plot for the Dorpers and Racka breeds. (**C**) Neighbor-joining tree for Dorpers and Racka sheep breeds.

**Figure 2 biology-12-00386-f002:**
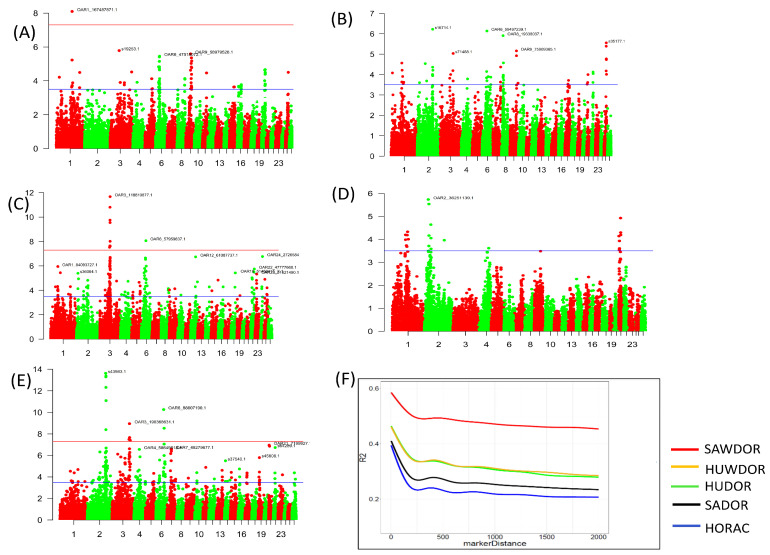
Genome-wide distribution of iHS values for (**A**)—HUDOR, (**B**)—HUWDOR, (**C**)—SADOR, (**D**)—SAWDOR, and (**E**)—HORAC sheep breeds. x-axis of Manhattan plots represents Autosome chromosome numbers, The red line represents the top 0.001 percentile distribution of iHS values, while the blue line represents the top 0.01 percentile, SNPs are annotated at *p* value of 0.00001. (**F**) Genome-wide linkage disequilibrium decay against marker distance in each population.

**Table 1 biology-12-00386-t001:** Basic population genomic statistics and runs of homozygosity.

FID	N	Ho ± SD	He ± SD	F ± SD	ROH (Mbps)				
					0–6	6–12	12–24	24–48	>48
SADOR	21	0.372 ± 0.164	0.368 ± 0.137	0.066 ± 0.041	994	193	51	6	2
HUDOR	20	0.365 ± 0.177	0.351 ± 0.145	0.173 ± 0.106	1386	293	74	20	4
HUWDOR	20	0.367 ± 0.164	0.363 ± 0.137	0.177 ± 0.064	1167	235	106	26	9
HORAC	20	0.376 ± 0.151	0.377 ± 0.130	0.046 ± 0.122	999	219	83	14	1

N = Number of samples; FID = Family identity; Ho = Observed heterozygosity; He = Expected heterozygosity; F = Average inbreeding coefficient; Mbps = Megabase pairs.

**Table 2 biology-12-00386-t002:** Population variation components and fixation index.

Source of Variation	d.f	Sum of Squares	Variance Components	Percentage of Variation
Among populations	3	79212.882	987.640 Va	9.76
Within population	132	773272.817	6467.576 Vb	90.24
Total	135	852485.699	10065.041	
Fixation index (FST) 0.097; *p*-value < 0.001				

## Data Availability

The raw data analyzed in this study is available upon request from the corresponding author.

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
