# Peer review of "Assessing the Genomics Structure of Dorper and White Dorper Variants, and Dorper Populations in South Africa and Hungary"

_biology, 2023, doi:10.3390/biology12030386_

Round 1
Reviewer 1 Report
Dear Authors,
my suggestions are in the attached file.

Author Response
The authors have prepared an interesting article on the different variants of the Dorper sheep breed. The Dorper breed was created for existence in the specific conditions of South Africa, but over time it gained popularity in other regions of the world. Therefore, the authors undertook genomic analyzes of several individuals from Hungary and South Africa.
Hence my remark - was the number of pieces not too small? About 15 individuals from Hungary were used. Results for the population with South Africa were taken from the database, with the final analyzes for SAWDOR omitted. I understand that an attempt has been made to investigate the genetic position of the Dorper breed within the global sheep population, however, the number of individuals tested has been relatively small to determine parameters for genetic diversity and to determine conservation strategies. Can you explain if this amount of sample did not affect the results?
Response
Thank you so much for your observation and valid concern. However, we believe that the samples used provide a representative genomic picture of the populations under study since we employed medium density SNP chips which have an adequate coverage across the genome. Secondly, we made a deliberate attempt to collect samples from unrelated individuals as much as possible. This was made possible by sampling from different farms as well as using farm records to select unrelated samples. We also worked within the recently published practical guide to genomic characterization of animal genetic resources by FAO (2023; https://doi.org/10.4060/cc3079en), in which the authors recommended at least 15 samples per population if the study utilizes markers along the whole genome. In our study, apart from white Dorper from South Africa (SAWDOR) which had 5 samples, all other samples were above 20. We therefore omitted SAWADOR in calculating the general statistics indices.
However, the article itself is structured in a correct and interesting way.
The experiment is appropriate and well planned, the bioinformatics analyzes are well chosen, the cited literature is relevant.
Below I present some comments.
Please consider correcting Obs Het, Exp Het for Ho, He throughout the text as such abbreviations are used in research articles.
Response
We appreciate and agree with your observation and as a result, we have corrected in the revised manuscript.
When you enter a parameter name with its abbreviation for the first time, use only the abbreviation in the rest of the text. Be consistent.
Response
We appreciate and agree with your observation and as a result, we have corrected in the revised manuscript.
In Table 1, explain what FID, Ho, He, SD mean and maybe better use the abbreviation ROH. In the table you have mixed abbreviations with full names. Might be better to unify the names. You also expanded the breed names in L 114-121. Consider whether in the table it is also necessary? Also, you have a pretty large standard deviation. Explain why? How does this affect the obtained results?
Response
We appreciate and agree with your observation and as a result, we have corrected in the revised manuscript.
We also agree that we had a large standard deviation of > 0.1. This is caused by several factors but in the present study, we hypothesize that, it could be due to the large genomic variation between individuals withing the population. This phenomenon is useful for breeders since it facilitates a long term within population selection. And therefore, we believe the large standard deviation could and did not affect our results.
Table 2 is missing a header. What is p?
Response
Thank you so much for your observation. P in this context meant p value. We have since expanded it in the manuscript.
Please check if the figures are correctly described. Figure 2 is missing a letter to the last figure (F).
Response
We appreciate and agree with your observation and as a result, we have corrected in the revised manuscript.
L227-L235: Ho's scores were almost always higher than He's (small exception - HORAC). Explain what this means for the genetic diversity of the tested sheep.
Response
Yes, it is true that Ho scores were slightly higher that He’s in all dorper populations. Although this scenario could be caused by many genomic forces including gene flow, mutation, non-random mating, selection among others, Dorper has been subjected to selection since its development, hence possibly this could have been the reason.
L300-L310: Could you explain how your results might affect implementations?
Response
Our results add information to the already existing information on the importance of maintaining genetic diversity in indigenous domestic animals’ species. It also reveals why Dorper breed has continued to be successful outside the development environments (South Africa) and also provides insight to the genomic regions associated with the adaptation of this breed.
Further, the results also reveal that White dorper and dorper (also known as black head dorper) are differentiated from each other and can easily be treated as different breeds, and their genetic management can therefore be improved.
L312-L320: Could the genomic differences between the Hungarian and African populations be due to the human factor?
Response
We believe that, although human mediated selection can also influence the genome architecture, environment has a huge influence especially if the animals are produced under uncontrolled environmental conditions. Human mediated selection is mainly driven by market demand (productivity) while environmental selection is mainly for survival. Therefore, in this case, the differences observed could be environmental considering the climatic differences between both regions.
Table S1 - prepare a table with data, because this information opens in excel in csv format.
Response
We appreciate and agree with your observation and as a result, we have prepared as recommended.
Figure S1 - no description for the figure.
Response
We appreciate and agree with your observation and as a result, we have corrected in the revised manuscript.
Reviewer 2 Report
The text is well written, the ideas organized and the data consistent for a publication. Good utilization of different software. After reading the manuscript, I present some considerations and suggestions.
I understand that the sample design is often adequate to the reality of collections, financing and projects. But a minimal sentence is in order to explain why the comparison between Hungary and South Africa is important. There is no logical relationship.
The results discussed do not respond to the objectives presented. Need to reorganize the thread
The title is very broad and touches on big points that are not completely covered by the content. There are no [conservation strategies] or [management practices] other than short sentences in the introduction and conclusions.
It is also about artificial selection. There is no reference about this. With this clear interference with offspring, it is to be expected that there is population structure.
To discuss signature of selection the Supplementary Table1 must be in the main text and discussions should be gene based. It is not clear what these selection signals actually are.
L84 - genomic modifications occur by chance mutations and by chance some cause phenotypic changes. Selection acts on phenotype and not [to adapt to the new environment].
L178 - in an attempt to write differently between Obs Het/ Exp Het it was so confusing that you have to read it several times to understand. It doesn't matter that both were high or low, what you have to do is discuss the result
L238 SADOR/SAWDOR are older than HUDOR/HUWDOR. The inbred relationship is not caused by sample size. If there is an inbred between DOR and WDOR, it is expected that the differential characteristics will be homogenized and if there are no other factors acting, the populations will be genetically more similar over the generations.
The very different sample sizes (200 and 48000) require statistical corrections. All calculated indices may be biased by unpaired sampling
L246 the number of genes change?? Can different animals of the same species lose/gain genes? Variants are about alleles and not about genes.
L303 [third-world countries] oh no please!!! is a disused expression associated with the classification of countries that positioned themselves as neutral during the Cold War. In addition to being outdated, it is quite prejudiced.
L314 [different selection pressures] again the text is elusive. Need to cite and discuss the different pressures
L319 is not exactly the popularity but the economic interest in sustaining such distribution
Fig2 I couldn't read it even by increasing it on the screen
Author Response
The text is well written, the ideas organized and the data consistent for a publication. Good utilization of different software. After reading the manuscript, I present some considerations and suggestions.
I understand that the sample design is often adequate to the reality of collections, financing and projects. But a minimal sentence is in order to explain why the comparison between Hungary and South Africa is important. There is no logical relationship.
Response
We appreciate and agree with your observation and as a result, we have corrected in the revised manuscript. We added the following hypothesis statement between lines 106 and 109: “We contrasted the genome architecture of the Dorper populations found in South Africa and Hungary because we hypothesize that the later may have undergone genomic evolution to survive the new temperate climate rather than the tropical climate in which they were originally developed.”
The results discussed do not respond to the objectives presented. Need to reorganize the thread
Response
We appreciate and agree with your observation and as a result, we have corrected in the revised manuscript. We reorganized our flow and shifted the excerpt between lines 95 and 104 to between lines 327 and 345. The section is the authors’ opinion on the approach developing countries could take to enhance genetic management of animal genetic resources.
The title is very broad and touches on big points that are not completely covered by the content. There are no [conservation strategies] or [management practices] other than short sentences in the introduction and conclusions.
Response
We appreciate and agree with your observation and as a result, we have reformulated the title to read “Assessing the genomics architecture of Dorper and White Dorper variants, and Dorper populations in South Africa and Hungary” in the revised manuscript.
It is also about artificial selection. There is no reference about this. With this clear interference with offspring, it is to be expected that there is population structure.
Response
We appreciate and agree with your observation; we agree that artificial selection also contributes to the population structure and can alter the allele frequencies within the population. However, artificial selection perse will tent to influence allele frequencies in all populations if they are in the same environment and if the selection is for the same traits, e.g., reproductive traits, growth etc., but mixing the matrix of artificial selection and survivability in different environments enhances the genomic variations.
To discuss signature of selection the Supplementary Table1 must be in the main text and discussions should be gene based. It is not clear what these selection signals actually are.
Response
We appreciate and agree with your observation and as a result, we have discussed selected signatures in the revised manuscript. However, we did not understand what the reviewer meant by “It is not clear what these selection signals actually are” statement.
L84 - genomic modifications occur by chance mutations and by chance some cause phenotypic changes. Selection acts on phenotype and not [to adapt to the new environment].
We appreciate and agree with your observation, in this regard, we believe that genomic modification acts on both phenotypes and adaptation to the new environment. Genomic modification changes the genome architecture; if a beneficial mutation occurred at a specific location, selection would favor this allele pattern and their frequency will increase in the population (and vice versa). Long term selection might lead the reduced diversity in the population at that genomic location. So, if the preferred pattern favored survival in that environment, it would exist in the population, with or without modifying the physical expression of the animals. And therefore, we say that, modified genomes might act on the phenotypes which is associated with adaptation, or might act on the adaptation without affecting the phenotypic expression.
L178 - in an attempt to write differently between Obs Het/ Exp Het it was so confusing that you have to read it several times to understand. It doesn't matter that both were high or low, what you have to do is discuss the result?
Response
We appreciate and agree with your observation and as a result, provided a discussion in the revised manuscript.
L238 SADOR/SAWDOR are older than HUDOR/HUWDOR. The inbred relationship is not caused by sample size. If there is an inbred between DOR and WDOR, it is expected that the differential characteristics will be homogenized and if there are no other factors acting, the populations will be genetically more similar over the generations.
Response
We appreciate and agree with your observation. We believe that, although only highest performing individuals are exported to other counties, leading to high levels of homozygosity at the locus of interest (which at times can be miss-interpreted as inbreeding by the genomic softwares), our conclusion was made based the long (>48 mega base pairs (mbps) runs of homozygosity which means, Dorper and White Dorper in Hungary separately might have a very recent common ancestor.
The very different sample sizes (200 and 48000) require statistical corrections. All calculated indices may be biased by unpaired sampling.
Response
Thank you so much for your observation and valid concern. However, we believe that the samples used provide a representative genomic picture of the populations under study since we employed medium density SNP chips which have an adequate coverage along the whole genome. Secondly, we made a deliberate attempt to collect samples from unrelated individuals as much as possible. This was made possible by sampling from different farms as well as using farm records to select unrelated individuals. We also worked within the recently published practical guide to genomic characterization of animal genetic resources by FAO (2023; https://doi.org/10.4060/cc3079en), in which the authors recommended at least 15 samples per population if the study utilizes markers covering the whole genome. In our study, apart from white Dorper from South Africa (SAWDOR) which had 5 samples, all other samples were more than 15. We therefore omitted SAWDOR in calculating the general statistics indices.
However, we included SAWDOR in other genomic evaluations to establish the relationship between breeds genomically. This study might not provide a well conclusive answer to the variations between dorper variants and sub-populations, but it now opens an opportunity for a worldwide genomic evaluation of Dorper and Dorper derived breeds.
L246 the number of genes change?? Can different animals of the same species lose/gain genes? Variants are about alleles and not about genes.
Response
Thank you for your question.
Yes, we believe that animals of the same species and breed in different production environment can gain or lose genes. This happens because animals in different environments are subjected to different environment-based selection pressures (e.g., temperature, humidity, altitude, pasture etc.). Favorable genes (including mutations) in that environment are selected upon while the unfavorable ones are selected against. In the long run, genes are gained or lost.
L303 [third-world countries] oh no please!!! is a disused expression associated with the classification of countries that positioned themselves as neutral during the Cold War. In addition to being outdated, it is quite prejudiced.
Response
We appreciate and agree with your observation and as a result, we revised the statement in the revised manuscript.
L314 [different selection pressures] again the text is elusive. Need to cite and discuss the different pressures
Response
We appreciate and agree with your observation and as a result, we rephrased the previous statement, We hope that the present phrase in the revised manuscript is more clearer.
L319 is not exactly the popularity but the economic interest in sustaining such distribution.
Response
We appreciate and agree with your observation. While agreeing with your suggestion, we also add that despite having economic interest, many countries imported the breed doe to their ability to perform in harsh environments but still produce as elite breeds.
Fig2 I couldn't read it even by increasing it on the screen
Response
We appreciate and agree with your observation and as a result, we have improved the resolution of the figure.
Round 2
Reviewer 1 Report
Dear authors, my comments are in the attachment.

Author Response
- Crossbreeds are written with the symbol '×' (for example: Dorper × Red Maasai). Correct throughout the text.
Response
We appreciate and agree with your observation and as a result, we have corrected in the revised manuscript.
- Please check if this part of the text is correct ‘Awassy Co. Farm first introduced the Dorper breed…..’
Response
We appreciate and agree with your observation. Our statement was made based on the information we found the research article published by Budai et al. (2013), and our extensive literature review has not successfully found any other contradicting information.
- L114: DNA extraction and genotyping of the Hungarian samples were genotyped…? I do not understand this sentence. DNA extraction genotyped? In addition, there is no description of the laboratory part. What did you do after DNA extraction? Unless the analyses were analogous to the cited work [19]. Think about whether you need to change this sentence/sentences.
Response
We appreciate and agree with your observation and as a result, we have changed the sentence to read: “DNA extraction and genotyping of the Hungarian samples was done by Neogen Company. Ovine50K Illumina microarray bead chips [19] were produced based on Oar v4.0 as the reference genome. We received genomic datasets inform of the final report from which the working datasets of Hungarian samples were generated.”
- L176: F = moment relatedness F coefficient. Are you sure it is correct?
Response
We appreciate and agree with your observation and as a result, we have corrected the statement to read “F = Average inbreeding coefficient”
- Consider discussing the number of used samples, why you got such a large standard deviation, why Ho was always greater than He. You've answered my questions, but isn't it worth mentioning to readers. I leave that to your consideration. It is not obligatory. If you think it's unnecessary, don't add it. This is your article.
Response
We agree with your observation and as a result, we have corrected discussed the results between lines 221 and 224 which reads “High standard deviation in dorper populations could be due to high genetic variation within the populations. Notably and expectedly, Ho scores in all populations except HORAC were higher than He possibly due to the selection process the breed has undergone since its development.”
Reviewer 2 Report
the modifications are satisfactory to result in a good publication
Author Response
the modifications are satisfactory to result in a good publication
Response
We are happy that the reviewer was satisfied with our responses and modifications made in the revised manuscript.